# High Surface Reactivity and Biocompatibility of Y_2_O_3_ NPs in Human MCF-7 Epithelial and HT-1080 Fibro-Blast Cells

**DOI:** 10.3390/molecules25051137

**Published:** 2020-03-03

**Authors:** Mohd Javed Akhtar, Maqusood Ahamed, Salman A. Alrokayan, Muthumareeswaran M. Ramamoorthy, ZabnAllah M. Alaizeri

**Affiliations:** 1King Abdullah Institute for Nanotechnology, King Saud University, Riyadh 11451, Saudi Arabiamramamoorthy@ksu.edu.sa (M.M.R.); 2Department of Biochemistry, College of Science, King Saud University, Riyadh 11451, Saudi Arabia; salrokayan@ksu.edu.sa; 3Department of Physics and Astronomy, College of Science, King Saud University, Riyadh 11451, Saudi Arabia; zabn1434@gmail.com

**Keywords:** surface adsorption, NP surface, biocompatibility, autophagy, nanomedicine, nanotoxicology

## Abstract

This study aimed to generate a comparative data on biological response of yttrium oxide nanoparticles (Y_2_O_3_ NPs) with the antioxidant CeO_2_ NPs and pro-oxidant ZnO NPs. Sizes of Y_2_O_3_ NPs were found to be in the range of 35±10 nm as measured by TEM and were larger from its hydrodynamic sizes in water (1004 ± 134 nm), PBS (3373 ± 249 nm), serum free culture media (1735 ± 305 nm) and complete culture media (542 ± 108 nm). Surface reactivity of Y_2_O_3_ NPs with bovine serum albumin (BSA) was found significantly higher than for CeO_2_ and ZnO NPs. The displacement studies clearly suggested that adsorption to either BSA, filtered serum or serum free media was quite stable, and was dependent on whichever component interacted first with the Y_2_O_3_ NPs. Enzyme mimetic activity, like that of CeO_2_ NPs, was not detected for the NPs of Y_2_O_3_ or ZnO. Cell viability measured by MTT and neutral red uptake (NRU) assays suggested Y_2_O_3_ NPs were not toxic in human breast carcinoma MCF-7 and fibroblast HT-1080 cells up to the concentration of 200 μg/mL for a 24 h treatment period. Oxidative stress markers suggested Y_2_O_3_ NPs to be tolerably non-oxidative and biocompatible. Moreover, mitochondrial potential determined by JC-1 as well as lysosomal activity determined by lysotracker (LTR) remained un-affected and intact due to Y_2_O_3_ and CeO_2_ NPs whereas, as expected, were significantly induced by ZnO NPs. Hoechst-PI dual staining clearly suggested apoptotic potential of only ZnO NPs. With high surface reactivity and biocompatibility, NPs of Y_2_O_3_ could be a promising agent in the field of nanomedicine.

## 1. Introduction

Screening of nanoparticles (NPs; defined as particles having, at least, one dimension below 100 nm) to be exploited in nanomedicine has been unprecedented in recent years since they provide multiple avenues in diagnosis and therapy of diseases. NPs could be engineered to target cancerous cells and release anticancer drugs therein, thus, avoiding normal cell toxicity that is a bigger obstacle in conventional chemotherapy [1,2]. NPs originating from lanthanide group exhibit remarkable properties owing to their high optical activity, surface reactivity and superior photo stability [3,4,5]. High refractory properties, good thermal conductivity, superior stability and excellent mechanical property make yttrium oxide (Y_2_O_3_) NPs a fascinating material [6]. Engineering NPs for active targeting to cancer cells while sparing normal cells, however, requires NPs with certain inherent properties. Obviously, biocompatibility and surface reactivity are some of the important criteria. Due to their intrinsic physical properties, inorganic NPs, especially of lanthanide group, offer highly tunable surface properties and stability over polymeric NPs [7,8]. NPs of antioxidant CeO_2_ and quite a different pro-oxidant (toxic) NPs of ZnO, for example, have received much attention for their potential usefulness in medicine though both NPs exhibit different biological effects [9]. Some investigators have reported that chemically related NPs of CeO_2_ and Y_2_O_3_ possess redox potential that enables them to mimic the activity of antioxidant enzymes such as superoxide dismutases (SODs) [10]. Moreover, as a novel treatment strategy, Y_2_O_3_ NPs could be used in the therapy of fulminant hepatic failure and oxidative stress-related diseases [11].

The present study was designed seeking a higher surface reactivity of NPs that could be exploited in nanomedicine. Since surface reactivity gives information about the extent to which NPs surface can be tuned, therefore, determining surface reactivity is considered a pre-requisite in the field of nanomedicine. Understanding the surface reactivity of protein molecules such as bovine serum albumin (BSA) is crucial towards predictability of drug interaction with a NP [12,13]. Quantification of BSA reactivity towards NPs have been one of the reliable methods in the evaluation of the degree of NPs activity towards biological models and molecules [14]. In this study, surface reactivity of BSA with Y_2_O_3_ NPs was studied and compared with the surface reactivity of CeO_2_ NPs and ZnO NPs. To the best of our knowledge, we could not find any report on BSA reactivity with that of Y_2_O_3_ NPs. There are, however, reports on BSA reactivity with the NPs of europium-doped Y_2_O_3_ (Y_2_O_3_:Eu^3+^) and ZnO NPs [14]. As will be discussed later, BSA affinity with Y_2_O_3_ NPs was found to be highest among all of the three NPs chosen in this study. Strong BSA reactivity suggested Y_2_O_3_ NPs could be a potential candidate in drug loading and delivery. In addition to BSA powder suspended in aqueous solution, surface reactivity of Y_2_O_3_ NPs towards culture media ingredient was also compared with that of CeO_2_ and ZnO NPs since surface modification can significantly reduce the toxicity of NPs [15].

Due to aforementioned properties, NPs of Y_2_O_3_, if found to be biocompatible, could serve the purpose of nano-carrier for various drugs. Biocompatibility of Y_2_O_3_ NP was examined in the two human cell lines- human breast (MCF-7) cells and fibroblast (HT-1080). These cells are a proven model for bio-medical testing [9,16,17]. This study, therefore, reports a comparative finding in terms of surface reactivity and biocompatibility of Y_2_O_3_ NPs with the chemically related NPs of CeO_2_ and chemically un-related NPs of ZnO. It is notable that NPs of Y_2_O_3_, like that of CeO_2_ NPs, have been reported to be antioxidant and protective in cells of neural [18,19] and retinal [20], whereas some investigators have reported this NP as oxidative and toxic [21,22]. In this study, we have provided a comparative data on surface reactivity with BSA and media components which is not known. Moreover, detailed mechanisms of potential toxicity carried out in the two human cell lines could further shed light on the discrepancy in toxicity level that exists for Y_2_O_3_ NP [18,19,20,21,22].

## 2. Results

### 2.1. Y_2_O_3_ NPs TEM Sizes, Hydrodynamic Sizes and Zeta Potential

NPs size calculated by TEM imaging at different resolution came to be 35 ± 10 nm while shapes appeared mostly cuboidal (Figure 1A,B). The matte texture observed in high-resolution (HR) TEM images (Figure 1C,D) confirms the crystallinity of Y_2_O_3_ NPs. 

Sizes and zeta potential of NPs in distilled water, phosphate buffer saline, serum free culture medium and complete culture media were measured to see the potential differences caused by serum and media components. Dynamic light scattering (DLS) measurements taken at 200 μg/mL show a differential dispersion of Y_2_O_3_ NPs in distilled water, phosphate buffer saline, serum free culture media and complete culture media as given in Table 1.

### 2.2. Y_2_O_3_ NPs Strongly React with BSA, Media and Non-Protein Serum Components

Surface reactivity of Y_2_O_3_ NPs with BSA was examined and compared with surface reactivity of two other NPs—CeO_2_ and ZnO—known to exert opposite responses in a wide variety of biological models. Among these NPs, Y_2_O_3_ NPs showed significantly higher BSA adsorption (Figure 2A). Control optical density of BSA (500 μg/mL in water) was taken as a reference point. BSA adsorption study with all three NPs was performed in water. BSA adsorbed, at 6 h of incubation, by NPs of Y_2_O_3_, CeO_2_, ZnO was found to be 330 ± 8, 145 ± 6 and 17 ± 4 μg, respectively. Similar is the trend at 24 h of incubation, though, with a slight decrease in BSA adsorption. To see the alteration in protein adsorption, if any, caused by proteins from other source (fetal bovine serum for example), or components of culture media, the Y_2_O_3_ NPs were suspended first in the liquid containing the respective adsorbent and incubated for 6 h or 24 h. After the respective period of incubation, NPs were centrifuged and in BSA containing water. Results clearly indicate that serum or media component caused significant shielding effect to BSA adsorption onto NPs surface (Figure 2B). Time dependent study shows that shielding effect due to serum or media component is quite stable. Data shows that Y_2_O_3_ NPs bind with serum proteins at rate of 30 ± 4 μg/mg NP when incubated for 6 h. BSA pre-treated NPs, however, did not show any noticeable amount of serum adsorption. Moreover, reactivity towards either BSA or media component is stable; neither BSA adsorption occurred to media treated Y_2_O_3_ NPs nor serum free media caused BSA displacement from Y_2_O_3_ NPs pre-treated with BSA (see summary in Table 2).

### 2.3. Y_2_O_3_ NPs Do Not Possess Inherent SOD-Like or CAT-Like Activity

Redox active NPs such as CeO_2_ are known to possess reactive oxygen species (ROS) scavenging capability. This ROS scavenging capacity plays critical role in the antioxidant activity reported of these NPs, though other factors might also exist [23,24]. In this study, SOD-like and CAT-like activity of Y_2_O_3_ NPs was measured along with NPs of CeO_2_ and ZnO. Table 3 shows that no significant enzyme-like activity of Y_2_O_3_ NPs was detected except for the NPs of CeO_2_.

### 2.4. Y_2_O_3_ Nps Did Not Cause Significant Decrease in Cell Viability 

There was no difference found in the cell viability data due to BSA treated or non-treated Y_2_O_3_ NPs in MCF-7 and HT-1080 cells (see Appendix A). The rest of the biological parameters were, therefore, conducted using naïve NPs of Y_2_O_3_, CeO_2_ and ZnO. As could be observed in Figure 3A–C, NPs of Y_2_O_3_ (and chemically similar CeO_2_ NPs at 100 μg/mL) did not cause inhibition to cell viability whereas ZnO caused 50% inhibition to cell viability (defined as IC50) at 41 ± 4 μg/mL to MCF-7 cells and 33 ± 4 μg/mL to HT-1080 cells.

### 2.5. Y_2_O_3_ NPs Did Not Induce Oxidative Stress 

Potential oxidative damage induced by NPs was assessed by cellular integrity directly affected by ROS. Lactate dehydrogenase (LDH) release, a marker of cell membrane damage, was measured in cells at concentrations of 50, 100 and 200 μg/mL exposed for 24 h (Figure 4A). Y_2_O_3_ NPs did not cause any LDH release in MCF-7 and HT-1080 cells at concentrations of 50, 100 and 200 μg/mL. Another marker of membrane damage, lipid peroxidation (LPO), followed similar trend in both cells (Figure 4B). It should be noted that statistically significance membrane damage caused by NPs of ZnO at IC50. As expected, antioxidant CeO_2_ NPs did not elevate markers of membrane damage. Similarly, Y_2_O_3_ NPs did not induce ROS that was measured by DCF probe and H_2_O_2_ specific sensor (Figure 4C,D). A standard oxidant, H_2_O_2_, was always included as a positive control of ROS during 2′, 7′-dichlorofluorescin diacetate (DCFH-DA) measurement (see Appendix A).

### 2.6. Y_2_O_3_ NPs Did Not Cause Mitochondrial Membrane Potential Induction 

To test further biocompatibility of Y_2_O_3_ NPs, mitochondrial membrane potential (MMP) was determined qualitatively and quantitatively using combined advantage of a fluorescence microscope (Leica DMi8, Wetzlar, Germany) and ImageJ software. Qualitative and quantitative results clearly suggest that Y_2_O_3_ NPs exposure at 200 μg/mL, highest concentration of NPs taken in this study, to both cells did not exert negative impact (Figure 5A,B). Punctate red fluorescence of JC-1 aggregate coming from intact mitochondria is a qualitative evidence of non-toxicity of Y_2_O_3_ NPs. In essence, there was no appreciable change in MMP after 24 h of incubation with any concentration of Y_2_O_3_ NPs relative to control cells.

### 2.7. Y_2_O_3_ NPs Did Not Cause Appreciable Change in Autophagy Activity 

Finally, a comparative potential of autophagy modulation due to the three NPs was evaluated to understand the mechanism of biocompatibility in greater depth because many biocompatible NPs established by cytotoxicity and oxidative stress later turned to be inducer of autophagy. Above mitochondrial data provide sufficient set of evidence that suggests Y_2_O_3_ NPs are biocompatible in MCF-7 and HT-1080 cells. Similar to the role of mitochondria in cell survival and death, lysosomes are thought to play critical part in determining the fate of the cells. The two related NPs-Y_2_O_3_ and CeO_2_, did not induce changes in lysosomal activity in MCF-7 (Figure 6A) and HT-1080 (Figure 6B) cells. LTR fluorescence, however, suggested a definite increase in lysosomal or acidic autolysosomes that occurred in ZnO NPs treated cells. A higher LTR fluorescence, quantified for both cells in Figure 6C, is equivalent to extensive acidic vesicle formation within cells. 

### 2.8. Y_2_O_3_ NPs Did Not Exhibit Apoptotic-Necrotic Potential

Hoechst and PI dual staining clearly shows NPs of Y_2_O_3_ and CeO_2_ to be non-cytotoxic (Figure 7). However, ZnO NP treated cell depicts IC50 toxicity. Moreover, an apoptotic mode of cells death in the case of ZnO NPs is evident by the fragmented and clumped pattern of chromatin condensation (see Appendix A).

## 3. Discussion

A variety of potential interacting component in relevant aqueous fluids may affect NPs native property prior to biological interaction [25,26]. Zeta potential and hydrodynamic sizes of NPs play crucial role in their secondary surface formation or coronization [27,28]. In this study, as determined by DLS, zeta potential of Y_2_O_3_ NPs suspension in complete culture media (−27.0 ± 1.2 mV) was much better than that in water (−16.0 ± 4.2 mV), PBS (−6.0 ± 2.4mV) and serum free culture media (−10.0 ± 4.0 mV). Interestingly, in a previous study, zeta potential of all aqueous (in deionized water) Y_2_O_3_ NPs preparations were highly negative with spherical Y_2_O_3_ NPs having an average zeta potential of −13 mV, platelets shaped NPs having an average zeta potential of −27 mV and the rod shaped NPs having an average zeta potential of −28 mV [29]. The respective mean hydrodynamic sizes for the spherical NPs (TEM size; 3 nm) was 700 nm, the platelet-shaped NPs (TEM size; 4 nm) was 700 nm and the rod-shaped NPs (TEM size; 11 nm) was 800 nm [29]. It should be recalled that in the present study the shape of the Y_2_O_3_ NP (TEM size; 35 ± 10 nm) was cubic in shape. In the present study, hydrodynamic size of Y_2_O_3_ NPs in water came out to be 1004 ± 134 nm while in complete culture media it was 542 ± 108 nm. The more negative zeta potential and lesser agglomeration observed in complete culture media for Y_2_O_3_ NPs is partially attributable to high surface reactivity of Y_2_O_3_ NPs with protein molecules as is evident by next experiment on Y_2_O_3_ NPs interaction with BSA proteins. In addition, BSA-treated Y_2_O_3_ NPs exhibited better dispersion in all of the tested aqueous media than native NPs (see Table 1). 

Protein adsorption to NPs is a dynamic process in fluids be it culture media, blood or other bio fluids independent of nature of the NPs either polymeric [28] or metallic [30]. One of the most notable alterations when NP is under biological fluids is the formation of NP-protein corona [28]. In addition to protein coating, there are reports about non-protein coating on Y_2_O_3_ NPs to achieve desirable outcome. Y_2_O_3_ NPs coating with glycerol citrate polymer, for example, caused increased antibacterial activity [31]. Realizing the importance of surface reactivity, in this study protein adsorption on to NPs in its native form was quantified using bovine serum albumin (BSA) in a time-dependent manner (see Figure 2). From the results, it can be safely concluded that the cellular response of Y_2_O_3_ NPs is actually the response of complete-media-component (that include serum proteins as well media ingredients) coated-Y_2_O_3_ NPs rather than only serum protein coated NPs. Thus, living systems mostly interact with the surface modified NPs rather than with bare NPs only when the NPs are surface reactive [32,33]. Bio-molecular corona formed on surface reactive NPs can provide stealth effect during NP uptake by immune cells [34]. Corona formation can also aid in biocompatibility [25]. Of the three NPs tested for surface reactivity, NPs of Y_2_O_3_ emerge with highest surface reactivity. The ROS scavenging capacity plays critical role in the antioxidant activity reported for CeO_2_ NPs [23,24]. Several reports have shown NPs of Y_2_O_3_ and CeO_2_ with similar characteristics to offer protection against induced toxicity by pro-oxidants [9,22]. Antioxidant enzyme-like activity exhibited by certain NPs such as that of CeO_2_ has been implicated in its potential protective effect against induced toxicity. Only CeO_2_ NPs, however, were found to exhibit SOD-like and catalase-like activity in this report (Table 3).

Cell viability in cells treated with the NPs of Y_2_O_3_ (35 ± 10 nm) and CeO_2_ up to the concentration of 200 μg/mL was find equal to that of control in the MCF-7 and HT-1080 cells. Some investigators, however, have found NPs of Y_2_O_3_ (41 ± 5 nm) to inhibit 50% cell growth (i.e., IC50) at the concentration of 108 μg/mL in human embryonic kidney (HEK293) cells [35]. Other researchers have examined NPs of Y_2_O_3_ and yttrium ions simultaneously in yeast and have concluded the yttrium ions derived from NPs responsible for the toxicity observed in yeast cells [36]. In mouse normal hepatocyte cell line, Y_2_O_3_ NPs (7–8 nm) coated with polymer of acrylic acid have been reported to cause toxicity at the concentration of 10 μg/mL [21]. On the other hand, NPs of Y_2_O_3_ have been reported to be antioxidant and protective in cells of neural [18,19] and retinal [20] origin. In this study, no significant difference in cell viability was observed for BSA-coated and native NPs of Y_2_O_3_ (See Appendix A). ROS such as H_2_O_2_ and O^2•−^, can damage cellular components when produced in large amounts but can initiate a diverse array of signaling pathways leading to control cell division, differentiation and migration [37]. Depending on its intracellular concentration and localization, H_2_O_2_ initiates either pro-apoptotic or anti-apoptotic response [38]. Levels of H_2_O_2_ in the range of 20–50 μM seem to have limited cytotoxicity in many cell types [39]. NPs of Y_2_O_3_ and CeO_2_ did not cause significant ROS induction in the two cell types. NPs of ZnO, however, induced H_2_O_2_ levels over 2-folds in both cells. MMP is the result of ROS that may contribute in inducing death activating pathways [38]. Therefore, NPs are reported to be strong modulator of MMP that can bring the onset of apoptosis, necrosis and autophagy [40,41]. Data suggested that there was no appreciable change in MMP after 24 h of incubation with any concentration of Y_2_O_3_ NPs relative to control cells. 

The tendency of autophagy modulation is considered as a survival strategy under short-term nutrient stress. However, if the depletion of antioxidants and nutrients is prolonged that might happen under toxicant exposure; the same autophagy may lead to course of cell death. Interestingly, NPs composed of rare earth oxide have been reported as a new class of autophagy inducers [42,43]. Certain NPs, as those of silica, have been widely used in nanomedicine and other applications due to their biocompatibility [44,45]. However, when examined on the parameter of autophagy, these NPs came to cause impairment in lysosomal activity and cellular protein clearance [46,47]. Given the importance of autophagy and lysosomal protein degradation in maintaining cellular homeostasis, therefore, in this report biocompatibility test was extended to analyzing the potential impact Y_2_O_3_ NPs might have on the activity of lysosomes. Recall that lysosomes are the major cellular organelle involved in the regulation of autophagy. In our study, the two related NPs-Y_2_O_3_ and CeO_2_, did not induce changes in lysosomal activity in both MCF-7 and HT-1080 cells. NPs of ZnO, used as toxic form of NPs, however, increased the number of cellular acidic vesicles and apoptotic mode of cell death in both cell types (see Appendix A for the apoptotic mode of cell death induced by ZnO NPs).

## 4. Materials and Methods 

### 4.1. Chemicals and Reagents

Fetal bovine serum, penicillin–streptomycin, Lysotracker Red-DND, CalceinAM were purchased from Thermo Fisher Scientific (Eugene, OR, USA). DMEM F-12K, Neutral red dye, MTT [3-(4,5-dimethyl thiazol-2-yl)-2,5-diphenyl tetrazolium bromide], Cacodylic acid, Pyrogallol, Bradford reagent, DCFH-DA, JC-1, H_2_O_2_, TBA, NADPH, pyruvic acid, H_2_O_2_ measuring kit, Hoechst, PI, all NPs were obtained from Sigma–Aldrich, USA. Ultrapure deionized-water was prepared using a Milli-Q system (Millipore, Bedford, MA, USA). All other chemicals used were of reagent grade.

### 4.2. Yttrium Oxide NPs

NPs of Y_2_O_3_ were commercially obtained from Sigma-Aldrich, USA. As per the information provided by supplier, TEM size of Y_2_O_3_ was below 50 nm. Color of nanopowder was white.

### 4.3. Transmission Electron Microscopy of Y_2_O_3_ NPs

Size of Y_2_O_3_ NPs was determined by field emission transmission electron microscopy (FETEM) (JEM-2100F, JEOL, Inc., Tokyo, Japan) using an accelerated voltage of 200 kV [48]. Normal and high resolution (HR) TEM images was also taken to observe the crystallinity of NPs as it happens in HR-TEM texture of typical crystals. Suspension of ultra-sonicated Y_2_O_3_ NPs was placed onto a carbon-coated copper grid, air dried and observed with FETEM. Purity of Y_2_O_3_ NPs was determined by energy dispersive spectrum (EDS) analysis.

### 4.4. Agglomeration and Zeta-Potential of Y_2_O_3_ NPs

Agglomeration behavior and zeta potential of Y_2_O_3_ NPs in water, phosphate buffer saline (PBS) and complete cell culture medium was determined by dynamic light scattering (DLS) system (Nano-ZetaSizer-HT, Malvern Instruments, Malvern, UK) as described by Murdock et al. [49]. NPs were freshly suspended in respective aqueous medium (200 µg/mL) and ultrasonicated for 10 min (Ultrasonic Cleaner-8891, Cole-Parmer, 625 Bunker Court Vernon Hills, IL USA). DLS measurement was carried out in cuvettes supplied with DLS system. For DLS measurement, a phenol red-free medium was used.

### 4.5. Determination of Surface Reactivity of NPs

To evaluate surface reactivity, NPs were treated with relevant chemicals such as bovine serum albumin (BSA), serum and media components. To determine the degree of protein adsorption, NPs and BSA were mixed in a 15 mL sterile tube. Final concentrations of BSA and NP were kept at 0.5 mg/mL and 1 mg/mL, respectively in a total volume of 10 mL. Then, a mixture of BSA and NP was put on a gentle shaker at room temperature after ultra-sonication for 20 min. From this NP-BSA mixture, 2 mL was taken in tubes and centrifuged for 10 min at 12,000× *g* at every 6 h and 24 h to measure protein adsorption. From the supernatant obtained, 100 µL each was taken from either NP-BSA mixture or BSA only solution, and added in cuvette containing 2.0 mL of Bradford reagent. A decrease in absorbance at 595 nm in NP-BSA mixture from BSA only solution was indicative of protein adsorption on NPs. Amount of BSA adsorbed on to NP surface was calculated using BSA standards at 595 nm and expressed as μg BSA adsorbed/mg of NP. Calculation method is similar to that of Patil et al. [50]. Protein adsorption to NPs is a dynamic process in liquids be it culture media, blood or other bio fluids. To see the alteration in protein adsorption, if any, caused by components of culture media, mixture of Y_2_O_3_ NPs was suspended in serum free (i.e., protein free) culture medium and left for 24 h. Then, BSA was added as above and any shielding effect caused by media ingredients in protein adsorption was measured at 6 h and 24 h as described above. Surface reactivity of Y_2_O_3_ NPs was also determined towards serum, a fluid containing diverse molecules such as albumin and other proteins, hormones, lipids, carbohydrates, etc.

### 4.6. Enzyme-Like Activity of NPs

Enzyme like activity of NPs was determined by the protocol of Marklund and Marklund [51] using autoxidation of pyrogallol. The principle of this method is based on the competition between the pyrogallol autoxidation by O_2_^•−^ and the dismutation of this radical by enzyme SOD in a 75 mM Tris/cacodylic acid buffer, pH 8.2. For uninhibited reaction, a 0.02 OD/min for 3 min at 420 nm was utilized. Amount of enzyme required to inhibit this reaction by 50% was defined as one-unit of SOD enzyme. Catalase was determined by recording the rate of H_2_O_2_ disappearance at 240 nm for 3 min as described by Aebi [52]. For this, a 100 mL phosphate buffer, pH 7.0 was mixed with H_2_O_2_ to give an absorbance of 0.54 at 240 nm. For calculation purposes, 0.436 mM extinction co-efficient for H_2_O_2_ at 240 nm was taken. Reaction was initiated by 50 μL of 1 mg/mL NPs suspension in one mL of phosphate buffer with H_2_O_2_ absorbance of 0.54 at 240 nm. A suitable NP control was used for subtracting absorbance decrease due to NPs settling in reaction cuvette from the absorbance caused by potential enzyme-like activity of NPs.

### 4.7. Cell Culture

Human breast cancer (MCF-7) cells and fibrosarcoma (HT-1080) cells (ATCC, Manassas, VA, USA) were maintained in DMEM-F12K supplemented with 10% fetal bovine serum, 100 U/mL penicillin and 100 µg/mL streptomycin, at 37 °C in a humidified 5% CO_2_ incubator (HERACell 150i, Thermo Fisher Scientific, Waltham, MA, USA). The cells were passaged for every 3–4 days before reaching confluence level.

### 4.8. Viability Studies

Cell viability was carried out by MTT, NRU and LDH assay as briefly described below. MTT assay was carried out according to the protocol described by Mosmann [53] with minor modifications. Briefly, around 20,000 MCF-7 cells per well were seeded in 96-well plates in a 100 µL of culture medium. Blue formazon formed in viable cells were solubilized and absorbance taken at 570 nm using a plate reader (Synergy HT, Bio-Tek, Winooski, VT, USA). Cell viability of treated and control cells is given as cell viability in percentage of control. To minimize the interference in absorption that may be potentially caused by the NPs, plates were centrifuged to suspend the NPs, and 100 µL supernatant was carefully transferred (without disturbing NPs at the plate bottom in the original experimental plate) to another fresh well in a 96-well plate as previously reported [54]. The neutral red uptake (NRU) assay is based on the method described by Repetto et al. [55]. Absorbance was taken at 540 nm using a plate reader (Synergy HT, Bio-Tek, Winooski, Vermont, US ) and data is given as cell viability in percentage of control. The activity of cytoplasmic LDH released into the culture media was determined with the method described [56]. A 100 µL sample from the centrifuged culture media was collected after the cells were treated for 24 h. The rate of NADH oxidation was determined by following the decrease in absorbance at 340 nm for 3 min at 30-s interval at 25 °C using a spectrophotometer (Genesys 10 Bio, Thermo Fisher Scientific, Madison, WI, USA). The amount of LDH released is represented as LDH activity (IU/L) in culture media.

### 4.9. Determination of Lipid Peroxidation

LPO was assessed by the thiobarbituric acid reactive substances (TBARS) assay, which detects mainly malondialdehyde (MDA), an end product of the peroxidation of polyunsaturated fatty acids and related esters. MDA was measured by slight modification of the method of Ohkawa et al. [57]. The absorbance of the supernatants was read at 532 nm. Results were calculated as nmol TBARS/mg of cellular protein using 1.56 × 10^5^ M^−1^cm^−1^ as molar extinction of MDA-TBA.

### 4.10. Measurement of Total Reactive Oxygen Species by DCFH-DA and H_2_O_2_ by Specific Sensor

The generation of intracellular ROS was measured using 2′, 7′-dichlorofluorescin diacetate (DCFH-DA) probe [58]. DCF fluorescent intensity was measured at emission from 528 nm band pass of plate reader (Synergy HT, Bio-Tek, Winooski, Vermont, USA). A specific H_2_O_2_ sensor was employed for the detection of intracellular H_2_O_2_ according manufacturer’s protocol (MAK164, Sigma-Aldrich, St. Louis, MO, USA). A standard of H_2_O_2_ was run and measured similarly using H_2_O_2_ sensor.

### 4.11. Ratio-Metric Measurement of Mitochondrial Membrane Potential by JC-1

Mitochondrial outer membrane potential (MOMP) in control and treated cells were determined by JC-1. In healthy cells with high mitochondrial membrane potential, JC-1 spontaneously forms complexes known as J-aggregates with intense red fluorescence. On the other hand, in apoptotic or unhealthy cells with low membrane potential, JC-1 remains in the monomeric form showing green fluorescence [59]. Cells were seeded at appropriate densities in 12-well plate. When treatment period was over, media was aspirated off from each well and labeled with 5 µM JC-1 in Hepes buffered HBSS for 20 min. Fluorescence pictures of JC-1 monomer and aggregate in cells were captured by a fluorescence microscope (DMi8, Leica Microsystems, Wetzlar, Germany, using objective 20×, Leica DFC450 C camera and its LAS software version 4.13).

### 4.12. Measurement of Lysosomal Activity by LTR

Lysosomal activity predominantly refers to intracellular acidic vesicles (lysosomes or fusion product of autophagosomes with lysosomes) during autophagy [60]. Lysosomal activity in cells can be tracked by LysoTracker (LTR) dyes. LTR probes freely permeate cell membranes and label live cells. One such LTR dyes is red fluorescent LysoTracker™ Red DND-99 (Thermo Fisher Scientific, Eugene, Oregon, US) as was used in our previous report [61]. After the specific treatment period, cells were labeled at the final concentration of 1 µM LTR within ongoing culture media. After 30 min of labeling in 12-well culture plates, cells were carefully washed with cold HBSS with three times to remove excess LTR. CalceinAM, a green fluorescence emitting live cell dye was labeled at 0.5 µM simultaneously in LTR imaging to corroborate live cells from dead ones. An increase in punctate red fluorescence of LTR in treated cells than that in control cells indicates an increase in lysosomes or intact acidic vesicles (autolysosomes) [62].

### 4.13. Apoptotic-Necrotic Potential Detection by Hoechst/PI Staining

A dual staining based on concurrent use of Hoechst and propidium iodide (PI) was carried out again to confirm the mechanism of toxicity if occurred by the NPs of Y_2_O_3_. This approach was based on the ability of Hoechst to label every cell including live as well as dead and ability of PI to penetrate only dead cells. Further, if there is toxicity, a fair idea about potential mechanism of cell death induced by either apoptosis or necrosis can be deduced by applying logical conventions set by many cytologists [63]. In apoptotic cells, most nuclei show clumped, compact and discrete fluorescence due the pattern of nuclear DNA breakdown whereas in necrotic cell, most nuclei show increased in perimeter, diffused and peripheral fluorescence. In control dead cells, nucleus is neither increased in perimeter (as in necrosis), nor show compactness and discrete fluorescence (as in apoptotic cells) but rather a smooth and centrally intense fluorescence [64]. This is an inexpensive and more dependable method which was successfully applied in our previous publication deciphering mode of cell death caused by apoptosis to that from apoptosis-independent [61].

### 4.14. Protein Estimation

The total protein concentration was measured by using a ready-to-use Bradford reagent with bovine serum albumin as protein standard [65].

### 4.15. Statistics

One way ANOVA (analysis of variance) followed by Dunnett’s multiple comparison tests were employed for statistical analysis of data. Statistical significance was attributed at *p* < 0.05. The experiment was repeated three times (*n* = 3) carried in triplicates. For a particular set of the experiment, a burst of images was captured at the constant exposure of time, gain, saturation and gamma. Images representing three independent experiments were used in this report. Calculation of corrected total cellular fluorescence (CTCF) was conducted in ImageJ software (NIH, Bethesda, MD, USA). Using the region of interest (ROI) manager command, the constant area selection was obtained to each image opened in ImageJ. CTCF was calculated by subtracting the mean of background (without cell) fluorescence from the mean of cellular fluorescence (i.e., mean integrated density). Scale bar in images was set using ImageJ after calibrating the scale in terms of pixels/micron according to the respective objectives used during imaging.

## 5. Conclusions

NPs of Y_2_O_3_ could be desired agent in drug loading and targeted delivery as shown by their high surface reactivity, biocompatibility and their inability to affect cellular autophagy functioning. Higher surface reactivity also provides opportunity to test drugs loading, and in the further fine tuning of the NPs surfaces with diverse array of available biocompatible surfactants [27].

## Figures and Tables

**Figure 1 molecules-25-01137-f001:**
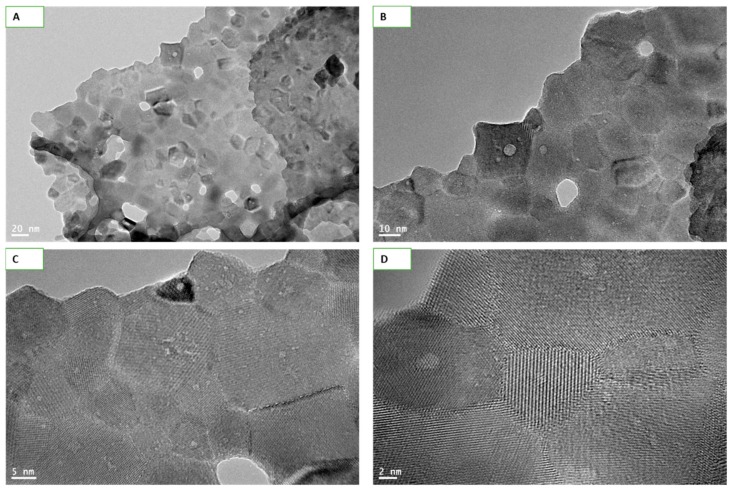
Transmission electron microscopy (TEM) images of Y_2_O_3_ nanoparticles (NPs) captured at different resolutions. Images taken at 20 nm (**A**) and 10 nm (**B**) clearly depicts the sizes and shapes of NPs. Size of NPs was calculated to be 35 ± 10 nm while shapes were mostly cuboidal. Matte texture, an evidence of crystallinity of the materials, was observed at the high resolutions of 5 nm (**C**) and 2 nm (**D**).

**Figure 2 molecules-25-01137-f002:**
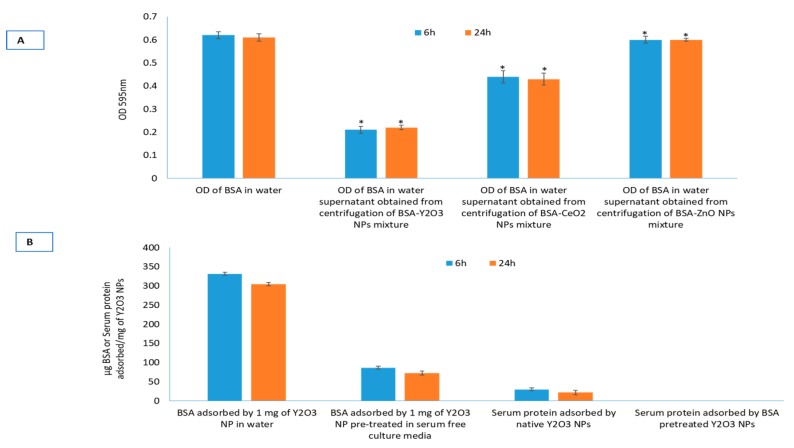
Surface reactivity of the three NPs with relevant biomolecules. Adsorption of bovine serum albumin (BSA) on to Y_2_O_3_, CeO_2_ and ZnO NPs (**A**). Adsorption to BSA was highest for Y_2_O_3_ NPs while least for ZnO NPs. Y_2_O_3_ NPs resulted in highest surface reactivity, therefore, displacement experiment was further extended to Y_2_O_3_ NPs to understand the relative stability of adsorption of BSA, serum albumin and media ingredients with respect to each other (**B**). Triplicates of each treatment group were used in each independent experiment. * Denotes a significant difference from the control (*p* < 0.05).

**Figure 3 molecules-25-01137-f003:**
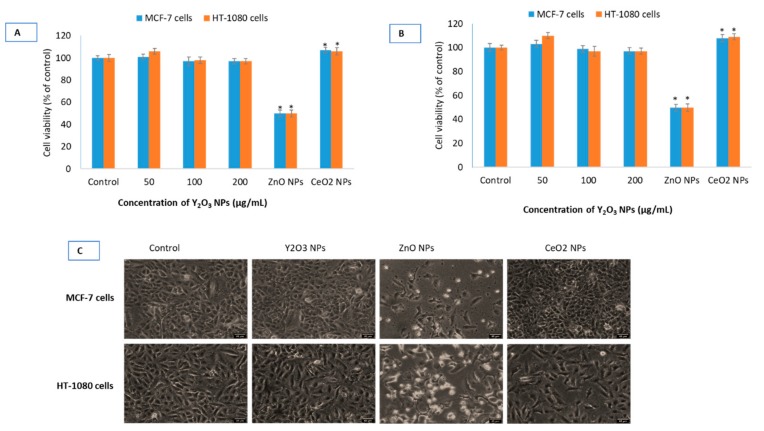
Cell viability in human breast (MCF-7) cells and human fibroblast (HT-1080) cells treated with the three NPs of Y_2_O_3_, CeO_2_ and ZnO determined by MTT (**A**) and neutral red uptake (NRU) (**B**) assay at indicated concentrations. It should be noted that a single concentration of CeO_2_ NPs (100 μg/mL) and ZnO NPs (IC50) have been used in all of the biological experiments since these two NPs have been taken as primarily for comparison purposes with that of Y_2_O_3_ NPs. Phase-contrast images (20×, Leica DMi8, Germany) of MCF-7 cells and HT-1080 cells treated with highest concentration (i.e., 200 μg/mL) of Y_2_O_3_ NPs, IC50 of ZnO and CeO_2_ (100 μg/mL) have been given (**C**). For IC50 calculation, a scatter plot in Microsoft Excel was inserted followed by setting the *Y*-axis to logarithmic. Then a trend line was selected and ‘exponential’ picked. Then ‘display equation’ was used in calculating ICs. IC50s calculations were further verified and confirmed from the online IC50 calculator (https://www.aatbio.com/tools/ic50-calculator) provided by AAT Bioquest, Inc. (CA 94085, USA). Scale-bar represents 40 μm (micron) in each image. Triplicates (*n* = 3) of each treatment group were used in each independent experiment. * Denotes a significant difference from the control (*p* < 0.05).

**Figure 4 molecules-25-01137-f004:**
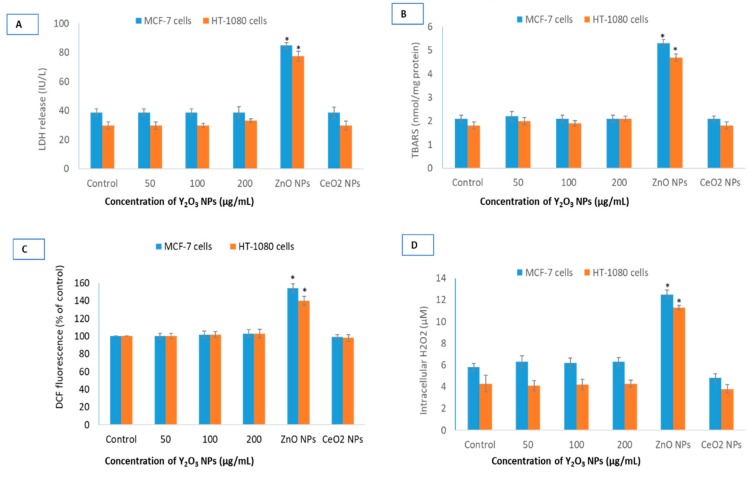
Comparative oxidative stress parameters for the three NPs. Potential lactate dehydrogenase (LDH) release (**A**), lipid peroxidation (LPO) induction (**B**), DCF (fluorescent part of the non-fluorescent DCFH-DA molecule that is formed inside cells after ROS induced cleavage) fluorescence (**C**) and intracellular H_2_O_2_ induction (**D**) in MCF-7 and HT-1080 cells by NPs of Y_2_O_3_, ZnO and CeO_2_. Triplicates (*n* = 3) of each treatment group were used in each independent experiment. * Denotes a significant difference from the control (*p* < 0.05).

**Figure 5 molecules-25-01137-f005:**
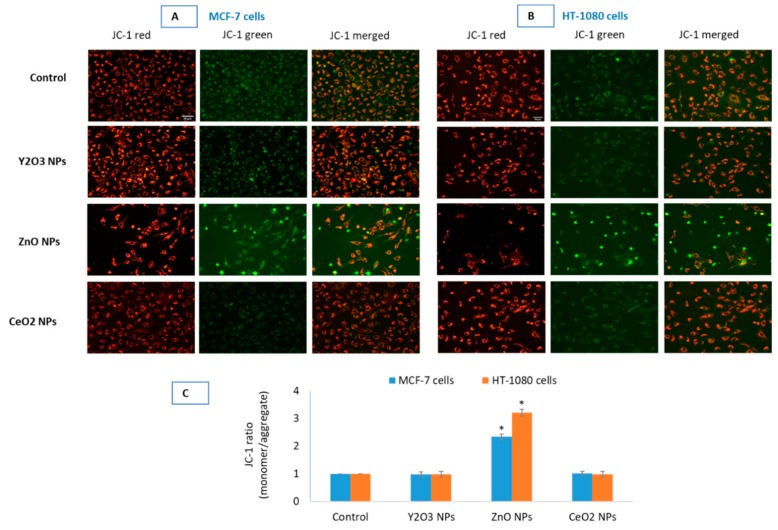
Comparative mitochondrial outer membrane potential (MOMP or simply MMP) was detected by JC-1 in control and in the cells treated with the NPs of Y_2_O_3_ (200 μg/mL), ZnO (IC50) and CeO_2_ (100 μg/mL) in MCF-7 (**A**) and HT-1080 (**B**). Images were captured in tandem for JC-1 aggregate (red) and JC-1 monomer (green). Quantification of MOMP (**C**) in the two cells is given as JC-1 ratio (monomer/aggregate) of corrected total cell fluorescence (CTCF) analyzed in open source ImageJ software from NIH. Fluorescence intensities and cell morphology suggest no significant induction of MOMP in cells treated with 200 μg/mL of Y_2_O_3_ NPs and CeO_2_ NPs with that of control cells. NPs of ZnO, as expected, induced significant MOMP in the two cells. Images (captured by Leica DFC450C camera fitted in Leica DMi8 microscope, Germany, using 20× objectives) are representative of the three independent experiments with similar results. Scale-bar represents 40 μm (micron) and given in the first image of each parameter. Data represented are mean ± SD of three identical experiments (*n* = 3). * statistically significant difference as compared to the controls (*p* < 0.05).

**Figure 6 molecules-25-01137-f006:**
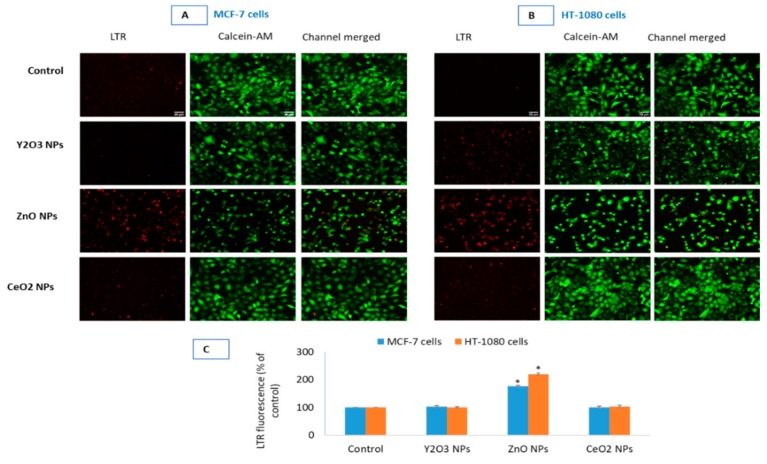
Lysosomal activity due to the NPs of Y_2_O_3_, ZnO and CeO_2_ was evaluated in MCF-7 cells (**A**) and HT-1080 cells (**B**). CalceinAM was used as control of live cell fluorescence. Quantification of LTR fluorescence (**C**) in the two cells is given as corrected total cell fluorescence (CTCF) of control cells vs. treated. CTCF was analyzed in open source ImageJ software from NIH. Triplicates (*n* = 3) of each treatment group were used in each independent experiment. * Denotes a significant difference from the control (*p* < 0.05).

**Figure 7 molecules-25-01137-f007:**
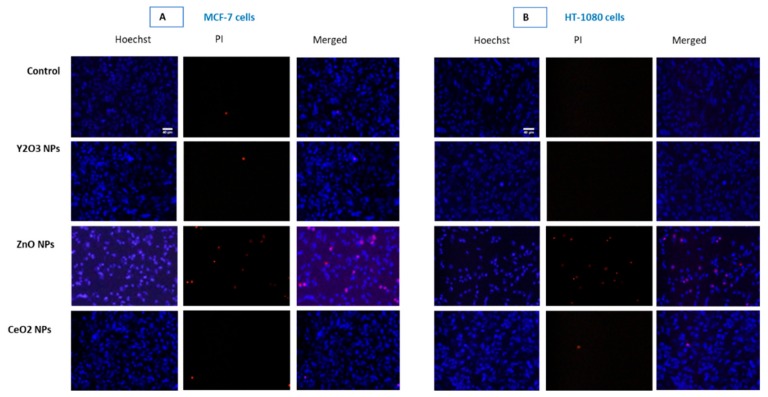
Propidium iodide (PI) staining clearly shows NPs of Y_2_O_3_ and CeO_2_ to be non-cytotoxic in MCF-7 cells (**A**) and HT-1080 cells (**B**). However, ZnO NP treated cell depicts IC50 toxicity. Moreover, an apoptotic mode of cells death in the case of ZnO NPs is evident by the fragmented and clumped pattern of chromatin condensation (see Appendix A). Triplicates (*n* = 3) of each treatment group were used in each independent experiment.

**Table 1 molecules-25-01137-t001:** Y_2_O_3_ NP’s properties in powder form and its behavior in relevant aqueous media.

Y_2_O_3_ NPs	
**Physico-Chemical Properties**	
TEM size	≤50 nm
Color	White
HR-TEM	Crystallite texture
TEM shape	Mostly cubic
EDS	Elemental impurities not detected
**Agglomeration and Zeta Potential in Aqueous Media**	
**Water**	
Hydrodynamic size	1004 ± 134 nm
Zeta potential	−16.0 ± 4.2 mV
Hydrodynamic size of BSA-treated NPs	350 ± 33 nm
Zeta potential of BSA-treated NPs	−33.0 ± 1.6 mV
**PBS**	
Hydrodynamic size	3373 ± 249 nm
Zeta potential	−6.0 ± 2.4mV
Hydrodynamic size of BSA-treated NPs	1479 ± 213 nm
Zeta potential of BSA-treated NPs	−13.0 ± 2.4mV
Serum free culture media	
Hydrodynamic size	1735 ± 305 nm
Zeta potential	−10.0 ± 4.0 mV
Hydrodynamic size of BSA-treated NPs	686 ± 142 nm
Zeta potential of BSA-treated NPs	−17.0 ± 4.0 mV
**Complete Culture Media**	
Hydrodynamic size	542 ± 108 nm
Zeta potential	−27.0 ± 1.2 mV
Hydrodynamic size of BSA-treated NPs	491 ± 89 nm
Zeta potential of BSA-treated NPs	−37.0 ± 1.2 mV

**Table 2 molecules-25-01137-t002:** Comparison of serum adsorption to NPs in media and serum with that of BSA adsorption to pristine and media treated NPs.

NPs	Serum Protein Adsorption in Complete Culture Media	Serum Protein Adsorption in Filtered Serum Only	BSA Adsorption to Pristine NPs in Water	BSA Adsorption to NPs Treated with Media Components
Y_2_O_3_	4.13 ± 0.5%	4.43 ± 0.5%	64.0 ± 1.7%	6.0 ± 1.5%
CeO_2_	2.3 ± 0.5%	2.3 ± 0.5%	28.0 ± 1.8%	3.0 ± 1.3%
ZnO	Not detected	Not detected	9.0 ± 1.6%	Not detected

**Table 3 molecules-25-01137-t003:** Reactive oxygen species (ROS) scavenging potential of NPs in respective buffer.

NPs	SOD-Like Activity	CAT-Like Activity
Y_2_O_3_	not detected	not detected
CeO_2_	detected significantly	detected significantly
ZnO	not detected	not detected

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
