# Peer review of "High Surface Reactivity and Biocompatibility of Y2O3 NPs in Human MCF-7 Epithelial and HT-1080 Fibro-Blast Cells"

_molecules, 2020, doi:10.3390/molecules25051137_

Round 1

Reviewer 1 Report

The aim of this study is to focus on global interactions of nanoparticles and biomolecules in biological environments, which play a critical role in the final applications in biomedicine.  The Authors failed to give convincing evidences of their broad revision of the recent literature indeed they are still missing some important issues.  Above all the Authors do not discuss the cell uptake and fate of the Y2O3 NPs. Moreover, as they are indicating Y2O3 NPs as promising delivery systems again the Authors do not give any comment on the load capacity of these Y2O3 NPs, nor any evidence of functionalization for enhancing the specific targeting. The Authors completely missed to investigate the biocompatibility of Y2O3 NPs in endothelial cells, missing an important milestone in term of the Y2O3  NPs feasibility in nanomedicine applications. The discussion need to be more oriented on the applicability of Y2O3 NPs in the field of the nanomedicine. Nothing about the load capacity of these NPs, or the surface properties dealing with specific functionalization orienteering the NPs targeting properties. Please try to better outline the novelty of the study providing a broader and deeper discussion of the recent advances in the field. The manuscript would need a thorough linguistic revision.

Reviewer 2 Report

The manuscript entitled “High surface reactivity and biocompatibility of Y2O3NPs in human MCF-7 epithelial and HT-1080 fibroblast cells” gives some interesting data concerning the biological response of Y2O3NPs  in two human cell lines. However, in my opinion, the manuscript needs major improvements before it can be published in  Molecules.

Major remarks:

1) Preparation of nanoparticles solution for characterisation need more precise description. How the nanoparticles were prepared for DLS measurement? Whether the nanoparticles solutions were prepared in the same way as in experiments with the cells? In what solvent the nanoparticles have been suspended?  If the solution of nanoparticles was prepared immediately before experiments or earlier?

2) DLS measurements need a more precise description. It is not clear how did the authors measure the size of nanoparticles and potential zeta (the type of cuvettes, pH of tested solutions etc). How long from the sonication, the size of NPs was measured? Whether the authors have studied the aggregation of nanoparticles in time? Y2O3NPs aggregate in nutrient solution and it can affect the phytotoxicity results.

Zeta potential is given without pH value what is unacceptable. Potential zeta obtained by authors without pH value cannot be compared to the results of the literature.

Authors gave the size of nanoparticles and zeta potential without standard deviations. If the results obtained by DLS were repeated or measurement was done only once? If measurements were repeated the size of Y2O3NPs and also potential zeta should be given with the standard deviations.

DLS measurements should not be taken in the complete medium because such measurement is unreliable (serum and the phenol red dye present in the medium interfere with the measurement).

3) As MTT and LDH are well known for the interference of particles, how has this been adjusted?

4) How was the IC50 calculated for ZnONPs?

5) Was there any positive control in DCFDA-measured ROS?

Round 2

Reviewer 1 Report

Still reamaining unsolved the issue of the NPs cell uptake studies and being only speculative the application of these NPs in cancer therapy, the authors updated the manuscript as they best following the reviewer suggestions.